# COVID-19 Vaccine Hesitancy in South Africa: Lessons for Future Pandemics

**DOI:** 10.3390/ijerph19116694

**Published:** 2022-05-30

**Authors:** Michelle Engelbrecht, Christo Heunis, Gladys Kigozi

**Affiliations:** Centre for Health Systems Research & Development, Faculty of the Humanities, University of the Free State, P.O. Box 339, Bloemfontein 9300, South Africa; heunisj@ufs.ac.za (C.H.); kigozign@ufs.ac.za (G.K.)

**Keywords:** COVID-19, vaccine hesitancy, vaccine literacy, health literacy, health behaviour, risk factors

## Abstract

Vaccine hesitancy, long considered a global health threat, poses a major barrier to effective roll-out of COVID-19 vaccination. With less than half (45%) of adult South Africans currently fully vaccinated, we identified factors affecting non-uptake of vaccination and vaccine hesitancy in order to identify key groups to be targeted when embarking upon COVID-19 vaccine promotion campaigns. A cross-sectional, anonymous online survey was undertaken among the South African adult population in September 2021. Our research identified race, interactive–critical vaccine literacy, trust in the government’s ability to roll out the COVID-19 vaccination programme, flu vaccination status and risk perception for COVID-19 infection as key factors influencing the uptake of COVID-19 vaccination. Respondents who did not trust in the government’s ability to roll out vaccination were almost 13 times more likely to be vaccine-hesitant compared to those respondents who did trust the government. Reliable, easy-to-understand information regarding the safety of COVID-19 vaccines is needed, but it is also important that vaccination promotion and communication strategies include broader trust-building measures to enhance South Africans’ trust in the government’s ability to roll out vaccination effectively and safely. This may also be the case in other countries where distrust in governments’ ability prevails.

## 1. Introduction

With the arrival of severe acute respiratory syndrome coronavirus 2 (SARS-CoV-2) in late 2019 and its rapid spread worldwide, scientists were hesitant to promise a quick vaccine. The fastest previous turnaround time for a vaccine’s development was for the mumps vaccine in the 1960s, which took four years to develop [1]. In the case of the COVID-19, the first vaccine, the Pfizer-BioNTech COVID-19 Vaccine, was approved for emergency use in about one year’s time [2]. This rapid progress, ostensibly without compromising the safety of the vaccine, was made possible by years of research on related viruses, the use of faster techniques, sufficient funding to run parallel clinical trials and accelerated approval from regulators [3]. This suggests that given the right circumstances and available resources, it may be possible to rapidly develop vaccines in the event of future pandemics, which are predicted to occur more frequently. Based on a study of a global dataset of historical epidemics from 1600 to the present, Marani et al., (2021) [4] calculated that one currently has about a 38% chance of experiencing an extreme pandemic such as COVID-19 in one’s lifetime, which could double in coming decades.

Following vaccine roll-out in 2021, there was a temporary reduction in COVID-19 cases, particularly in countries with sufficient vaccine stocks and effective promotional campaigns [5]. Currently (22 May 2022), 65.7% of the world’s population has received at least one dose of a COVID-19 vaccine [6], with African countries generally having a much smaller percentage of their population vaccinated than more developed countries [7]. Just 15.9% of people in low-income countries have received at least one dose [6]. As of 10 May 2022, 45% of South African adults 18 years and older were fully vaccinated (meaning that they had received a Johnson & Johnson Vaccine or Pfizer first and second dose) [8], which was far less than the set goal to fully vaccinate 70% of the population by the end of 2021 [9]. Recent developments in the country allude to the possibility of mandatory vaccination, with suggested amendments to the National Health Act relating to the surveillance and control of notifiable medical conditions: “Any person with a confirmed or suspected case of a notifiable medical condition may not refuse to submit to mandatory prophylaxis, treatment, isolation or quarantine in order to prevent transmission” [10]. At present, the only prophylaxis available for COVID-19 is vaccines.

Since the development of the first vaccines, it was not uncommon for people to be hesitant to vaccinate. In 2019, vaccine hesitancy (i.e., the unwillingness or refusal to vaccinate despite the availability of vaccines) was listed as one of the top ten threats to global health, with the World Health Organization (WHO) reporting that vaccines prevent 2–3 million deaths per year and that a further 1.5 million deaths could be averted if global vaccine coverage improved [11]. A systematic review of factors that contribute to vaccine hesitancy and acceptance during previous pandemics/epidemics—influenza A/H1N1 pandemic and Ebola virus disease—identified the following: demographic factors including race, age, sex, education and employment; costs; access; risk perception; trust in health authorities; safety and efficacy of the vaccine; and (mis)information about the vaccine [12]. Research on COVID-19 vaccine hesitancy identified similar reasons for not wanting to vaccinate: demographic factors including age [13,14], gender [14], race [14,15,16], education [14,15] and income [15,17]; conspiracy theories and misinformation [13,17,18,19]; concerns about vaccine safety [13,20,21,22,23] and side effects [17,19]; distrust of the government [17]; poor health literacy [24]; lack of access to an online vaccination registration platform [17]; and not taking the seasonal flu vaccine [13].

Following a review investigating the extent and determinants of COVID-19 vaccine hesitancy in South Africa, the authors concluded that although a number of factors influence COVID-19 uptake, more research is needed for definitive conclusions to be drawn and appropriate targeted strategies need to be developed that focus on sub-groups more prone to vaccine hesitancy [25]. With this in mind, our research aimed to identify factors associated with non-uptake of a COVID-19 vaccine as well as vaccine hesitancy in order to identify key groups to be targeted when embarking upon COVID-19 vaccination promotional campaigns.

## 2. Materials and Methods

### 2.1. Design, Setting and Sample

South Africa’s national vaccination programme commenced on 17 February 2021 and followed a staggered approach with healthcare workers being the first to be vaccinated, followed by different age groups. Vaccination is voluntary and free of charge. Persons are required to register on the national registration portal, and after registering they can go directly to their preferred vaccination site or make use of a booking system. At the time of the study (September 2021), vaccinations for persons 18 years and older had just become available. A cross-sectional, anonymous online survey was undertaken among the South African adult population (i.e., persons 18 years and older) in September 2021. At this point, slightly more than one-fifth of the adult population was vaccinated against COVID-19 [9], and the country was experiencing the tail-end of the third wave of COVID-19 infections, driven by the Delta variant.

The survey was advertised on multiple platforms, including word-of-mouth, social media (i.e., the University of the Free State Facebook and Twitter accounts) and the data-free Moya mobile messaging application [26], which has more than five million subscribers. In total, 10,466 adults participated in the survey, with slightly less than half (n = 5000) accessing the survey via the Moya application, while the remainder were exposed to social media advertisements or word-of-mouth (n = 5466).

### 2.2. Research Instrument and Data Collection

The online questionnaire was available in seven of the most frequently spoken South African languages—Afrikaans, English, Sesotho, Setswana, Tsepedi, Xhosa and Zulu—and comprised the following sections: socio-demographic information, including gender, age, race, education and employment status; health and COVID-19; uptake of COVID-19 vaccines; and the standardised Health Literacy about Vaccination in Adulthood (HLVa) Scale [27]. The scale comprises 12 items and two sub-scales measuring functional and interactive–critical vaccine literacy (VL), respectively. Functional VL refers to language capabilities, such as basic reading skills and comprehension of read content. Thereupon, interactive–critical VL focuses on more advanced cognitive skills, such as problem solving and decision making. Four-point Likert scales were used to rate the responses for the items measuring functional VL (4—never, 3—rarely, 2—sometimes, 1—often) and interactive–critical VL (1—never, 2—rarely, 3—sometimes, 4—often). Previous research found that the HLVa Scale has suitable psychometric characteristics for the subjective measure of VL in individual and population studies [28,29]. A cut-off value of ≤2.50 was proposed for limited VL [28].

The online questionnaire was hosted on a data-free website which was open for one month (September 2021). Potential respondents were informed that the questionnaire would take about ten minutes to complete and was anonymous and that they could select the language that they wished to complete the questionnaire in. They were also required to indicate that it was the first time that they were participating in the survey before they could access the information leaflet and informed consent documents. Only after the informed consent form was completed could participants proceed to answer the questionnaire.

### 2.3. Data Analysis

Data was analysed in IBM SPSS version 27 [30]. The data were described using frequency counts and percentages for categorical variables and means and standard deviations for continuous variables. Binomial logistic regression was used to determine which factors were significantly associated with not having a COVID-19 vaccine as well as vaccine hesitancy. Vaccine hesitancy was measured by asking respondents who had not yet vaccinated if they would do so in the future. Respondents who indicated “No” or who were “Unsure” were categorised as vaccine-hesitant. All assumptions for binomial logistic regression were met. Independent variables included in the models were: firstly, age (34 years and younger, 35 years and older; the reason for this classification was that the survey was undertaken in September 2021, which was also the month that South Africa opened up COVID-19 vaccination for all persons 34 years and younger); secondly, gender (male and female, other was dropped as there were only 11 respondents in this category); thirdly, race (African/Black, Coloured, White; due to the low number of respondents who self-identified as Asian, this category was not included in the regression model); fourthly, education (no formal, primary school, secondary school, tertiary); fifthly, employment status (employed, unemployed); functional VL (higher, limited); interactive–critical VL (higher, limited); and, sixthly, trust in the government’s ability to roll out the COVID-19 vaccination programme (yes, no). Two further variables were added to the model predicting COVID-19 vaccine hesitancy: firstly, flu vaccination status (yes, no) and, secondly, perceived risk for COVID-19 infection (yes, no/not sure).

### 2.4. Ethics

Ethical clearance was obtained from the Health Sciences Research Ethics Committee (HSREC) at the University of the Free State (UFS-HSD2021/0750/3108). Participation in the study was voluntary. The potential respondents were provided with sufficient information regarding the study in order to make an informed choice on whether or not to participate. Data were secured in encrypted files. Information was available for respondents requiring vaccine-related information as well as the contact details for free counselling services should they experience any emotional distress due to COVID-19.

## 3. Results

### 3.1. Demographic and Background Information

Table 1 indicates that approximately two-thirds of the respondents were male (65.1%) and self-identified as African/Black (63.4%). The average age was 34.5 years (SD 13.679). More than half of the respondents had a secondary education (57.5%), and 40.7% had a tertiary education. Three out of five (60.4%) respondents were unemployed.

### 3.2. Health and COVID-19

Overall, 47.8% of respondents described their health as being very good/excellent, while 34.7% indicated being in good health (Table 2). The majority of respondents (76.4%) had not taken the annual flu vaccine. Three-quarters of respondents had not had COVID-19, and only 9.7% thought that they were at risk for becoming infected with COVID-19. The most frequently mentioned practice that respondents engaged in to prevent infection with COVID-19 was to wear a face mask in public spaces (20.0%). It is, however, concerning to note that key COVID-19 infection control practices were not engaged in by large proportions of the respondents (i.e., wearing a mask 49%; regularly washing hands with soap and water 57.4%; avoiding big groups 60.5%; and regularly sanitising 67.4%).

### 3.3. COVID-19 Vaccination

Two out of five respondents (40.0%) had taken a COVID-19 vaccine. The main reasons for having the vaccine included: to protect against infection (27.3%); believing that there was no harm in taking a vaccine (16.3%); and thinking that the benefits of having a vaccine outweighed the risks (14.8%). The 60.0% of respondents who had not had a COVID-19 vaccine provided the following reasons: concerns about side effects (26.1%); the vaccines had been developed and approved too rapidly to be trusted (12.6%); and not trusting the government (11.8%). About three out of ten respondents who had not been vaccinated (29.2%) were unsure if they would in the future, while 15.4% said that they would not (see Table 3).

### 3.4. Factors Associated with Non-Uptake of COVID-19 Vaccination

The multivariate logistic regression model for not having had a COVID-19 vaccine (Table 4) was statistically significant, implying that the predictors as a set reliably distinguished between persons who had taken the COVID-19 vaccine and those who had not (X^2^ (10) = 3348.185, *p* < 0.001). The model explained 37.5% (Nagelkerke R2) of the variance in the tendency to not take a COVID-19 vaccine and correctly classified 75.7% of the cases. After controlling for other variables in the model, all eight predictor variables were found to be statistically significant (*p* < 0.05): age, gender, race, education, employment, functional VL, critical–interactive VL and trust in the government’s ability to roll out the COVID-19 vaccination programme. Persons 34 years and younger were 2.9 times more likely not to be vaccinated than persons 35 years and older (AOR = 2.870, *p* < 0.001). Females (AOR = 1.106, *p* = 0.047) were more likely than males not to be vaccinated. Compared to persons who self-identified as White, those who self-identified as African/Black (AOR = 6.002, *p* < 0.001) or Coloured (AOR = 4.856, *p* < 0.001) were 6.0 times and 4.9 times, respectively, more likely not to have been vaccinated. Compared to persons with a tertiary education, persons with no formal education/primary education (AOR = 2.337, p<0.001) and secondary education (AOR = 1.724, *p* < 0.001) were 2.3 times and 1.7 times, respectively, more likely not to be vaccinated. Unemployed persons were 1.6 times more likely than employed persons not to be vaccinated (AOR = 1.658, *p* < 0.001). Persons with limited functional (AOR = 1.114, *p* = 0.31) and limited interactive–critical (AOR = 1.345, *p* = 0.002) VL were more likely not to be vaccinated than persons with higher levels of functional and interactive–critical VL. Persons who did not trust in the government’s ability to roll out the vaccine programme were five times more likely not to be vaccinated than persons who did trust in the government’s ability to do this (AOR = 5.090, *p* < 0.001).

### 3.5. Factors Associated with COVID-19 Vaccine Hesitancy

The multivariate logistic regression model for COVID-19 vaccine hesitancy (Table 5) was statistically significant, implying that the predictors as a set reliably distinguished between persons who would take the COVID-19 vaccine and persons who would not (X^2^ (12) = 2017.937, *p* < 0.001). The model explained 40.5% (Nagelkerke R2) of the variance in the tendency to be hesitant to take COVID-19 vaccines and correctly classified 78.3% of the cases. After controlling for other variables in the model, five predictor variables were found to be statistically significant (*p* < 0.05): race, interactive–critical VL, trust in the government’s ability to roll out the COVID-19 vaccination programme, flu vaccination status and risk perception for COVID-19 infection. Compared to respondents who self-identified as White, respondents who self-identified as African/Black (OR = 0.158, *p* < 0.001) or Coloured (OR = 0.238, *p* < 0.001) were less likely to be vaccine-hesitant. Persons with limited interactive–critical VL were approximately two times more likely to be vaccine-hesitant compared to persons with higher levels of interactive–critical VL (OR = 1.974, *p* < 0.001). Persons who did not trust the government’s ability to roll out the vaccine programme were 13.1 times more likely to be vaccine-hesitant than persons who did trust in the government’s ability to do this (OR = 13.057, *p* < 0.001). Persons who had not had the annual flu vaccine were 1.7 times more likely to be hesitant to have a COVID-19 vaccine than persons who had received the flu vaccine (OR = 1.663, *p* < 0.001). Persons who perceived themselves not to be at risk for COVID-19 infection were almost 2.0 times as likely to be vaccine-hesitant compared to persons who did perceive themselves to be at risk for infection (OR = 1.973, *p* < 0.001).

## 4. Discussion

Our research identified factors associated with uptake of COVID-19 vaccination as well as vaccine hesitancy so as to identify key groups to target when promoting vaccines. While there have been other South African studies investigating vaccine intent (including hesitancy) and predictors thereof [17,23,31], ours is one of the first studies to collect data at a time when all adult citizens were eligible for a COVID-19 vaccine. Vaccine intent as measured by two earlier studies in 2021 (both using the same database) found that the intent to accept the COVID-19 vaccine among the adult population in South Africa increased from 71% in February/March 2021 to 76% in April/May 2021 [23,31], while Katoto et al., (2022) [17] reported that 32% of their respondents were vaccine-hesitant in a study undertaken during June/July 2021. Our research, conducted a few months later in September 2021, found that three out of five respondents had not vaccinated against COVID-19, and of these respondents, 44.6% were hesitant (either said no or were unsure) about whether they would vaccinate in future, which is slightly higher than that reported in the above-mentioned studies. This suggests that the intention to receive the COVID-19 vaccine is fluid, with acceptance rates varying at different points in time. This was also illustrated in international studies. In China, for example, researchers found that the intention to immediately accept vaccination after a COVID-19 vaccine became available declined substantially, from 52.2% in March 2020 to 24.7% in November–December 2020, reportedly due to concerns about vaccine safety [22]. Similarly, longitudinal research in Kuwait [32] and the US [33] also reported a decline in vaccine acceptance over time. This evidence aligns with the Sage Working Group’s (2014) [34] (p. 7) elucidation that, “Vaccine hesitancy is complex and context specific, varying across time, place and vaccines. It is influenced by factors such as complacency, convenience and confidence.”

The main reason for not having had a COVID-19 vaccine, or for being unsure of whether to have a vaccine, was related to concerns about possible side effects of the vaccine. This finding is in line with international [35], regional African [36] and local South African [17,23] studies conducted since the roll-out of COVID-19 vaccine programmes. Even while COVID-19 vaccines were still under development, fear of possible side effects was cited as a key reason for not having a vaccine once it became available [13,14,37], indicating that risk perception is a major barrier to uptake of COVID-19 vaccination. This relates to the second most frequently cited reason for not having a COVID-19 vaccine observed in the current study, namely, the belief that that these vaccines had been developed too rapidly to be trusted. This suggests that the public needs to understand how it was possible to develop COVID-19 vaccines so rapidly while still ensuring the safety of the vaccines. Such information should be communicated simply in order to aid understanding in a sea of information and misinformation circulating about COVID-19 vaccines [38].

Our study, supported by similar research, highlights that the following key factors need to be taken into consideration when targeting vaccine hesitancy: race [14,15,16]; interactive–critical vaccine literacy [27,29]; trust in the government’s ability to roll out the COVID-19 vaccination programme [17]; flu vaccination status [13]; and risk perception for COVID-19 infection [39]. With regard to race, compared to persons who self-identified as White, persons who self-identified as African/Black or Coloured were less likely to have had a COVID-19 vaccine, but were more likely to be in favour of having a COVID 19 vaccine in the future. To the contrary, Nguyen et al., (2022) [16], in a longitudinal study in the United States of America (USA) and United Kingdom (UK), found that Black participants were less likely to have had a COVID-19 vaccine and more likely to remain vaccine-hesitant.

Vaccine literacy, a relatively under-researched area, has been found to be associated with COVID-19 vaccine uptake. As with Biasio et al., (2020) [27] in an Italian study, and Gusar et al., (2021) [29] in a Croatian study, we found that persons who had limited interactive–critical vaccine literacy scores were also more likely to be vaccine-hesitant. Ratzan (2011) [40] notes that the concept of VL does not just denote knowledge about vaccines, but also refers to the (health) system’s ability to communicate clear and easy-to-understand information about vaccines. This aligns with another one of our findings, that the fear of side effects prevented people from taking a COVID-19 vaccine, again illustrating the need for information regarding COVID-19 vaccines that readers can easily understand, engage with and use to make decisions. Authors such as Gisondi et al., (2022) [38] have raised concerns that the focus has largely been on developing vaccines but not on distributing reliable and easy-to-understand information about the vaccines. The Organisation for Economic Co-operation and Development (OECD) has emphasised the role of governments in enhancing public trust in COVID-19 vaccination [41].

The significance of health literacy, and by extension VL, was highlighted in a Turkish study, where health literacy was found to play a mediating role between distrust of the healthcare system and vaccine hesitancy [42]. This is an important lesson from our study, where we found that more than a quarter of our respondents distrusted the government’s ability to roll out the COVID-19 vaccine programme. Furthermore, our respondents, who did not trust in the government’s ability to roll out vaccination, were 13 times more likely to be vaccine-hesitant compared to those respondents who did trust the government. This lack of trust is not without due cause; South African investigators flagged COVID-19 contracts worth around ZAR 2.1 billion (USD 137.12 million) for possible corruption and fraud [43]. Stemming from this investigation, 224 government officials underwent disciplinary action, 386 people were referred to the National Prosecuting Authority, and 330 companies were recommended for blacklisting [44]. Distrust of the South African government’s ability to roll out the COVID-19 vaccine programme was also reported by other local studies [17] as one of the reasons why people would not vaccinate. Contrarily, Burger et al., (2021) [23] reported that less than 5% of their respondents attributed a lack of trust in the South African government as a reason for vaccine hesitancy. Nevertheless, the message is clear: the public needs to have faith in the government’s ability to roll out the COVID-19 vaccine programme. Practical guidelines in this regard include open and honest communication with the public, prosecution of corrupt officials and seeing role models take up COVID-19 vaccination.

The South African Medical Council (2022) [45] notes that vaccine hesitancy was evident in South Africa long before COVID-19, and highlights that in 2009, national and provincial health managers were concerned about achieving optimal child vaccination coverage in the country. Vaccine hesitancy was also evident with the roll-out of the human papillomavirus (HPV) vaccine in public schools to grade 4 girls, aged 9 years [46]. Even annual flu vaccination uptake is low in the country, with only about 5% of persons with private medical insurance having taken the flu vaccine in 2015 [47]. The corresponding figure in our 2021 study was 23.6%. We also found that persons who had not had the flu vaccine were more likely to display COVID-19 vaccine hesitancy. Vaccine-hesitant patterns of behaviour have been observed in countries as diverse as Jordan [13], Turkey [42], Malta [20], the UK [14,16], the USA [15,16,21,35] and China [22]. The COVID-19 pandemic thus provides a unique opportunity to address vaccine hesitancy trends, not just in South Africa [17,23,25], but around the world. Research suggests that to be effective, COVID-19 communication strategies should focus on “broader trust-building measures that focus on relationships, transparency, participation, and justice” [25] (p. 921).

The value of our study is that it builds on a growing database of research investigating COVID-19 vaccine hesitancy. Ours is the first study to be undertaken in South Africa at a time when all adults older than 18 years of age qualified for a COVID-19 vaccine and therefore provides insights into hesitant behaviours when a vaccine is readily available. As with all research, ours too has limitations. The use of an online survey to collect our data excluded members of the population who do not own a smart phone, tablet or computer. As most respondents were unemployed (n = 6324, 60.4%) and because we did not collect information on the value of the grants they received (if any), we decided not to include income in the analysis. Additionally, as the sample was not randomly selected, caution needs to be exercised when interpreting and generalising the results. Furthermore, the cross-sectional nature of the data does not allow for interpretation of causality. As the data are self-reported, we are reliant on the participants’ honesty. Finally, as this was a self-administered online questionnaire, we were cognizant of the length of the questionnaire and the time it would take to complete. As such, we were limited in the number of questions that could be included.

## 5. Conclusions

Our research identified race, interactive–critical vaccine literacy, trust in the government’s ability to roll out the COVID-19 vaccination programme, flu vaccination status and risk perception for COVID-19 infection as key factors influencing the uptake of COVID-19 vaccines. The findings suggest that groups to target in vaccine promotion efforts include: those who self-identify as White, have lower levels of critical–interactive VL skills, distrust the government’s ability to roll out the vaccine programme, have not had the flu vaccine and do not see themselves as being at risk for COVID-19 infection. It is clear that reliable, easy-to-understand information regarding the safety of COVID-19 vaccines is needed, but it is also important that vaccination promotion and communication strategies include broader trust-building measures to enhance South Africans’ trust in the government’s ability to roll out COVID-19 vaccination effectively and safely. This may well apply in other countries where lack of trust in the government’s ability to do this prevails.

## Figures and Tables

**Table 1 ijerph-19-06694-t001:** Demographic and background variables.

Variable		n	%
Gender (N = 10,466)	Male	6812	65.1
	Female	3643	34.8
	Other	11	0.1
Age (N = 10,466)	34 years and younger 35 years and older	6501 3965	62.1 37.9
Race (N = 10,466)	African/Black	6636	63.4
	White	2493	23.8
	Coloured *	1190	11.4
	Asian	147	1.4
Education (N = 10,466)	No formal education	54	0.5
	Primary school	130	1.2
	Secondary school	6023	57.5
	Tertiary education	4259	40.7
Employment status (N = 10,466)	Unemployed	6324	60.4
	Employed full-time	2393	22.9
	Employed part-time	999	9.5
	Retired	608	5.8
	Student	142	1.4

* Mixed ethnic descent.

**Table 2 ijerph-19-06694-t002:** Health and COVID-19.

Variable		n	%
Current health (N = 10,466)	Excellent	2508	24.0
	Very good	2491	23.8
	Good	3643	34,7
	Fair	1477	14.1
	Poor	347	3.3
Flu vaccine (N = 10,466)	Had the flu vaccine	2468	23.6
	Did not have the flu vaccine	7998	76.4
COVID-19 (N = 10,466)	Had COVID-19	1559	14.9
	Did not have COVID-19	7930	75.8
	Not sure if had COVID-19	977	9.3
Perceived risk for COVID-19 (N = 8917)	At risk for COVID-19	867	9.7
	Not at risk for COVID-19	5171	58.0
	Not sure if at risk for COVID-19	2879	32.3
Protective behaviour (N = 27,457) *	Wearing a face mask in public spaces	5328	20.0
	Regularly washing hands with soap and water	4461	16.7
	Avoiding big groups	4131	15.5
	Avoiding close contact with others	4042	15.2
	Regularly sanitising	3414	12.8
	Staying at home more	3166	11.9
	Regularly cleaning home surfaces	2096	7.9

* Total number of responses—all applicable responses could be selected.

**Table 3 ijerph-19-06694-t003:** COVID-19 vaccination.

Variable		n	%
COVID-19 vaccination (N = 10,465)	Had a COVID-19 vaccine	4190	40.0
	Had not had a COVID-19 vaccine	6275	60.0
Reasons for taking a COVID-19 vaccine (N = 5834)	To protect against infection	1591	27.3
	There is no harm in having a COVID-19 vaccine	952	16.3
	Benefits of taking a COVID-19 vaccine outweigh the risks	866	14.8
	It is available for free	791	13.6
	It will help eradicate infection	722	12.4
	Many people have had a COVID-19 vaccine	296	5.1
	Doctor recommended a COVID-19 vaccine	231	4.0
	Well-known/respected people have taken a COVID-19 vaccine	177	3.0
	There is sufficient evidence regarding the safety and efficacy of COVID-19 vaccines	159	2.7
	Forced to for work/travel	49	0.8
Reasons for not being sure/not taking a COVID-19 vaccine (N = 25,816)	Concerns about side-effects	6732	26.1
	COVID-19 vaccines were developed and approved too rapidly to be trusted	3240	12.6
	Not trusting the government	3052	11.8
	Fear of needles	2728	10.6
	Prefer to acquire natural immunity	2656	10.3
	Don’t think it will be effective	2008	7.8
	Against vaccines in general	1836	7.1
	Not at risk for COVID-19	1600	6.2
	COVID-19 vaccines are promoted for commercial gains of pharmaceutical companies	1176	4.6
	Don’t have time to go for a vaccine	540	2.1
	Conspiracy theories—reduce the population/kill people	84	0.3
	Pregnant/breastfeeding	68	0.3
	Having co-morbidities	56	0.2
	Religious beliefs	40	0.1
Vaccination intentions (N = 6275) *	Intend to have a COVID-19 vaccine	3473	55.3
	Do not intend to have a COVID-19 vaccine	969	15.4
	Not sure about having COVID-19 vaccine	1833	29.2
Vaccine literacy (N = 10,466X)	Higher functional VL Limited functional VL Higher interactive–critical VL Lower interactive–critical VL	6254 4212 9608 858	59.8 40.2 91.8 8.2

* Excludes respondents already vaccinated.

**Table 4 ijerph-19-06694-t004:** Factors associated with non-uptake of COVID-19 vaccination.

Variable		Unadjusted Odds Ratio (95% CI)	Adjusted Odds Ratio (95% CI)
Age	35 years and older (ref)		
	34 years and younger	5.611 (5.144–6.120)	2.864 (2.568–3.193)
Gender	Male (ref)		
	Female	1.599 (1.469–1.741)	1.118 (1.011–1.236)
Race	White (ref)		
	African/Black	9.502 (8.523–10.594)	6.056 (5.193–7.062)
	Coloured	7.033 (6.034–8.197)	4.913 (4.078–5.918)
Education	Tertiary (ref)		
	Secondary	2.987 (2.750–3.244)	1.724 (1.556–1.910)
	Primary/no formal	3.822 (2.706–5.398)	2.337 (1.587–3.441)
Employment	Employed (ref)		
	Unemployed	3.176 (2.916–3.460)	1.658 (1.491–1.844)
Functional VL	Higher functional VL (ref)		
	Limited functional VL	1.735 (1.598–1.884)	1.114 (1.010–1.229)
Interactive–critical VL	Higher interactive–critical VL (ref)		
	Limited interactive–critical VL	1.898 (1.622–2.222)	1.345 (1.118–1.619)
Government’s ability to roll out vaccines	Trust government (ref)		
	Do not trust government	1.863 (1.698–2.044)	5.090 (4.454–5.817)

**Table 5 ijerph-19-06694-t005:** Factors associated with COVID-19 vaccine hesitancy.

Variable		Unadjusted Odds Ratio (95% CI)	Adjusted Odds Ratio (95% CI)
Age	35 years and older (ref)		
	34 years and younger	0.530 (0.469–0.598)	1.085 (0.908–1.296)
Gender	Male (ref)		
	Female	0.992 (0.895–1.099)	0.986 (0.863–1.128)
Race	White (ref)		
	African/Black	0.095 (0.074–0.122)	0.158 (0.110–0.228)
	Coloured	0.160 (0.121–0.213)	0.238 (0.160–0.353)
Education	Tertiary (ref)		
	Secondary	0.632 (0.567–0.706)	0.914 (0.788–1.060)
	Primary/no formal	0.650 (0.457–0.924)	0.706 (0.453–1.100)
Employment	Employed (ref)		
	Unemployed	0.596 (0.529–0.672)	0.903 (0.763–1.070)
Functional VL	Higher functional VL (ref)		
	Limited functional VL	0.864 (0.765–0.936)	0.938 (0.823–1.070)
Interactive–critical VL	Higher interactive–critical VL (ref)		
	Limited interactive–critical VL	2.418 (2.035–2.872)	1.974 (1.584–2.459)
Government’s ability to roll out vaccines	Trust government (ref)		
	Do not trust government	15.665 (13.629–18.006)	13.507 (11.212–15.207)
Flu vaccine	Had flu vaccine (ref)		
	Not had the flu vaccine	1.772 (1.462–2.149)	1.663 (1.298–2.219)
Risk for COVID-19	At risk for COVID-19 (ref)		
	Not at risk for COVID-19	2.017 (1.642–2.477)	1.973 (1.529–2.545)

## Data Availability

Data supporting reported results can be requested from the first author.

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
