# Peer review of "COVID-19 Vaccine Hesitancy in South Africa: Lessons for Future Pandemics"

_ijerph, 2022, doi:10.3390/ijerph19116694_

Round 1
Reviewer 1 Report
How come income was not studied? Could there have been a different on vaccine hesitancy due to income? Or on perceived health status (e.g. those that feel healthy vs. those feeling unhealthy).
Author Response
Dear Reviewer 1, thank you for your feedback. We have addressed your suggestions as follows:
|
Comment |
Response |
|
How come income was not studied? Could there have been a different on vaccine hesitancy due to income? |
Income was not included because about six in every ten (n=6,234, 60.4%) respondents were unemployed and we lacked information on the value of the grants they received, if any. This limitation of the study is now acknowledged in the Discussion (lines 335-337). |
|
Or on perceived health status (e.g. those that feel healthy vs. those feeling unhealthy). |
Perceived health status (current health) was included (line 168). |
Reviewer 2 Report
The manuscript represents an interesting way of monitoring the general population's response to the vaccines. The study, in general, is well defined; however, there are several issues
1.-in the text there is a difference between virtual and oral questionnaires, they should be comparable and validated.
2.-the results for the Asian volunteers, odds ratio, generate a comparison with other races is not correct due to the low number of individuals. Please refer to this population only in the text or in a separate table.
3.-There are more people willing to be vaccinated than vaccinated (Table 3) why? How is the vaccination program, it required appointments, and it is easily accessible? At the time of the study, how many people had received both vaccinations? and the time span? In some underdeveloped countries, there are individuals whose second dose was more than two months after the first dose.
4.-Is there any question that could be used to monitor if the vaccinated person was able to convince people to be vaccinated? The study seems to be focused only on individual perception. How many of the participants could be from the same family?
5. What is the recommendation of the authors for future pandemics?
Author Response
Dear Reviewer 2
Thank you for your feedback. We have addressed your suggestions as follows:
|
Comment |
Response |
|
1. -in the text there is a difference between virtual and oral questionnaires, they should be comparable and validated. |
There is no mention of oral questionnaires in the manuscript. To make sure that this is not misunderstood we added the word ‘online’ in line 99. |
|
2.-the results for the Asian volunteers, odds ratio, generate a comparison with other races is not correct due to the low number of individuals. Please refer to this population only in the text or in a separate table. |
Due to their under-representation (n=147, 1.4%), and after consultation with a biostatistician, respondents identifying as Asian, are now excluded from the regression analyses (Tables 4 and 5) and related text. |
|
3.-There are more people willing to be vaccinated than vaccinated (Table 3) why?
How is the vaccination program, It required appointments, and it is easily accessible?
At the time of the study, how many people had received both vaccinations? and the time span? In some underdeveloped countries, there are individuals whose second dose was more than two months after the first dose. |
Only those respondents who were not vaccinated (n=6,275) were asked about their vaccine intentions. In the background section we note that even by March 2022, less than half (45%) of South African adults were fully vaccinated (lines 45-46).
Information on the accessibility of the vaccination programme, the requirement for registration and the option of an appointment has been added (lines 82-86).
The reason why our study conducted in September 2022 did not include second-dose vaccination, was that second-dose J&J vaccination only became available in South Africa in February 2022. Furthermore, as different age groups were became eligible for vaccination at different points in time (lines 82-86), not everyone would have already qualified for a second Pfizer vaccination. |
|
4.-Is there any question that could be used to monitor if the vaccinated person was able to convince people to be vaccinated?
The study seems to be focused only on individual perception.
How many of the participants could be from the same family? |
No, due to the restriction on the number of questions in a questionnaire designed to take about 10 minutes to complete, our questionnaire did not address this issue.
Yes, the study is limited to individuals’ perceptions.
Because this was an anonymous online survey, we do not know how many participants were from the same family. |
Reviewer 3 Report
Comment to authors
Abstract
Good but I would like to say that the authors can give a better suggestion in the conclusion part, something real and valuable and at the same time accessible to the country under study, which is also a practical suggestion for other countries.
Keywords
Please use more related MESH terms, at least use 5 keywords. My suggestion: Health Behavior; risk factors; Health Literacy.
Introduction
Is there a source for this comment? Please cite: “This suggests that given the right circumstances and available resources, it may be possible to rapidly develop vaccines in the event of future pandemics, which are predicted to occur more frequently”
I would recommend to the author to give more data about the factors evaluated in the mentioned study (I suggest displaying influential risk factors with numbers, if possible): “A systematic review of factors that contribute to vaccine hesitancy and acceptance during previous pandemics/epidemics – influenza A/H1N1 pandemic and Ebola Virus Disease – identified the following: demographic factors including race, age, sex, education, and employment; costs; access; risk perception; trust in health authorities; safety and efficacy of the vaccine; and (mis)information about the vaccine [12].”
Materials and Methods
Research instrument and data collection
Were there any questions about how much information people had about the vaccine? Interested in injecting a specific type of vaccine?
How was the questionnaire survey how people were concerned about the safety of the vaccine?
This information is also important related to vaccine hesitancy and concerns about vaccine safety; have they been reviewed? Depending on the country of manufacture, type of vaccine platform, and free access to vaccine clinical information.
How long did it take to answer the questionnaire?
Most Socio-demographic variables include age, sex, education, migration background and ethnicity, religious affiliation, marital status, household, employment, and income. Related to them, I think some are important. For example, religious affiliation and income are very impressive, were they checked?
Has anyone been asked about the extent of coercion to get vaccinated or not? In traditional societies, especially among married people, this is an influential factor.
Please provide a summary of the extraction of data. Who and how many people have evaluated the quality of answers and extracted the data for analysis?
Results
Table 3. COVID-19 vaccination: What was the type of answer to the questions related to vaccine injection or non-injection? Could the participants answer or select several options at the same time? If yes, the number of simultaneous choices of several options in the second group (Reasons for not being sure/not taking a COVID-19 vaccine, N=25,816) was much higher than in the first group (Reasons for taking a COVID-19 vaccine, N=5,834). How is the unification of the answers done? Has sufficient and equal explanation and attention been paid to both vaccinated and non-vaccinated people? Most vaccinated people give only one reason?
Best regards
Author Response
Dear Reviewer 3
Thank you for your feedback. We have addressed your suggestions as follows:
|
Comment |
Response |
|
Abstract Good but I would like to say that the authors can give a better suggestion in the conclusion part, something real and valuable and at the same time accessible to the country under study, which is also a practical suggestion for other countries. |
We have adapted the conclusion in the abstract (lines 19-23) as well as in the main text (lines 351-357) in an attempt to improve the suggestions, holding possible value for other countries as well. We now suggest that the need to enhance South Africans trust in the government’s ability to safely and effectively roll out COVID-19 vaccination may also be applicable to citizens of other countries where issues of trust in vaccination prevail. This change to the conclusion is justified by the following sentence in the discussion section (lines 291-293): “The Organisation for Economic Co-operation and Development (OECD) has emphasised the role of governments in enhancing public trust in COVID-19 vaccination [41].” Note that suggestions that can be considered ‘practical’ are also made in the Discussion section (lines 311-313): “Practical guidelines in this regard include open and honest communication with the public, prosecution of corrupt officials, and seeing role models take up COVID-19 vaccination.” |
|
Key words: Please use more related MESH terms, at least use 5 keywords. My suggestion: Health Behavior; risk factors; Health Literacy. |
Five key words are now indicated: COVID-19; vaccine hesitancy; health literacy; health behaviour; risk factors (lines 24-25). |
|
Introduction Is there a source for this comment? Please cite: “This suggests that given the right circumstances and available resources, it may be possible to rapidly develop vaccines in the event of future pandemics, which are predicted to occur more frequently.” |
No reference for this statement is indicated because it is inferred from the above sentences and cited literature (lines 31-36). |
|
I would recommend to the author to give more data about the factors evaluated in the mentioned study (I suggest displaying influential risk factors with numbers, if possible): “A systematic review of factors that contribute to vaccine hesitancy and acceptance during previous pandemics/epidemics – influenza A/H1N1 pandemic and Ebola Virus Disease – identified the following: demographic factors including race, age, sex, education, and employment; costs; access; risk perception; trust in health authorities; safety and efficacy of the vaccine; and (mis)information about the vaccine [12].” |
The source of this information is a systematic review (Truong et al. 2022). Numbers and other statistics are not provided for the individual risk factors. |
|
Materials and Methods Research instrument and data collection Were there any questions about how much information people had about the vaccine? Interested in injecting a specific type of vaccine? |
Due to the restriction on the number of questions in a 10-minute questionnaire, we could not include questions on how much information about the vaccine people had or whether they were interested in receiving a particular vaccine. |
|
How was the questionnaire survey how people were concerned about the safety of the vaccine? This information is also important related to vaccine hesitancy and concerns about vaccine safety; have they been reviewed? Depending on the country of manufacture, type of vaccine platform, and free access to vaccine clinical information. |
We established respondents’ reasons for having undergone/not having undergone vaccination and report the numbers that thought there was sufficient evidence regarding the safety and efficacy of COVID-19 vaccines, those with concerns about side-effects and those who thought COVID-19 vaccines were developed and approved too rapidly to be trusted in Table 3 (line 180). |
|
How long did it take to answer the questionnaire? |
Ten minutes. This is now mentioned in the Research instrument and data collection section (lines 115-116). |
|
Most Socio-demographic variables include age, sex, education, migration background and ethnicity, religious affiliation, marital status, household, employment, and income. Related to them, I think some are important. For example, religious affiliation and income are very impressive, were they checked? |
In terms of religion, almost three quarters (n=7,801; 74.6%) of respondents indicated their religion as Christian. Religion was thus excluded from the analysis.
Income was not included because about six in every ten (n=6,234, 60.4%) respondents were unemployed and we lacked information on the value of the grants they received, if any. This limitation of the study is now acknowledged in the Discussion section (lines 335-337). |
|
Has anyone been asked about the extent of coercion to get vaccinated or not? In traditional societies, especially among married people, this is an influential factor. |
Due to the restriction on the number of questions that could be asked, this variable was not investigated. |
|
Please provide a summary of the extraction of data. Who and how many people have evaluated the quality of answers and extracted the data for analysis? |
The online questionnaire ensured data completeness by disallowing progression without an answer to each question. The first author and an experienced data manager, Ms Bridget Smit – cf. Acknowledgements (lines 372-373), checked the quality of the answers and transferred the data from the online questionnaires to SPSS for analysis. |
|
Results Table 3. COVID-19 vaccination: What was the type of answer to the questions related to vaccine injection or non-injection? Could the participants answer or select several options at the same time? If yes, the number of simultaneous choices of several options in the second group (Reasons for not being sure/not taking a COVID-19 vaccine, N=25,816) was much higher than in the first group (Reasons for taking a COVID-19 vaccine, N=5,834). How is the unification of the answers done? Has sufficient and equal explanation and attention been paid to both vaccinated and non-vaccinated people? Most vaccinated people give only one reason? |
At the time of our study (September 2021), non-injectable alternatives were still in development (https://www.news24.com/health24/medical/infectious-diseases/coronavirus/fear-of-needles-keeping-you-from-covid-19-vaccine-researchers-testing-other-options-20211001).
The respondents could provide as many reasons as they liked for having or not having a COVID-19 vaccination. Multiple responses were analysed in SPSS using the “Restructure” function. |
Reviewer 4 Report
The paper is well written and structured very well according to the rules of writing a scientific paper. The different elements of methodology have been presented transparently. The research results have been discussed interestingly and relate to a significant topic. The work contains an interesting discussion and conclusions, which take the form of practical considerations.
To make the paper even better, I suggest the authors include the following comments:
Line 42, 45: "fully vaccinated" - does this mean they have two doses of mRNA vaccine or two/one dose of vector vaccine? Does this mean they are also vaccinated with a booster dose?
Line 81: I think it would be helpful to add a brief note at this point regarding when vaccination started in South Africa and whether all population groups were covered - what seems obvious to the authors is not understood by readers from other countries because the vaccination schedule was different in different countries - this is also important in the context of what the authors write about in lines 121-122
Line 145, in Table 1, the incorrect category LGBTIQ was used in the Gender category - it is worth appreciating the researchers' good intentions to include other genders besides male over female. However, in the future, it should be remembered that LGBTIQ is not a gender, as lesbians can be both female and gays can be male. "other" or "intersex" should be used as the third answer in such cases.
Line 145 Table 1. The authors write in the Data analysis that they use age as an independent variable, but there is no age distribution of the respondents on demographics in the table. Was this intentional?
Line 148: wrongly summed results should be 57.8%, not 27.8%
Line 16-161: Is this a good place for this sentence? Perhaps it should be under the table or in Table 1?
Line167-168: it will be better "for three out of ten people" (29.2%)
Author Response
Dear Reviewer 4
Thank you for your feedback. We have addressed your suggestions as follows:
|
Comment |
Response |
|
Line 42, 45: "fully vaccinated" - does this mean they have two doses of mRNA vaccine or two/one dose of vector vaccine? Does this mean they are also vaccinated with a booster dose? |
‘Fully vaccinated’ in the South African context is now explained as having received a Johnson & Johnson Vaccine or Pfizer first and second dose (lines 46-47). Second-dose J&J vaccine was only available from 21 February 2022 (https://www.nicd.ac.za/covid-19-vaccine-booster-shot-frequently-asked-questions/). |
|
Line 81: I think it would be helpful to add a brief note at this point regarding when vaccination started in South Africa and whether all population groups were covered - what seems obvious to the authors is not understood by readers from other countries because the vaccination schedule was different in different countries - this is also important in the context of what the authors write about in lines 121-122. |
The starting point of the national vaccination programme, 17 February 2021, is now indicated as well as the fact that a staggered approach was followed to vaccinate the population (line 81-82) |
|
Line 145, in Table 1, the incorrect category LGBTIQ was used in the Gender category - it is worth appreciating the researchers' good intentions to include other genders besides male over female. However, in the future, it should be remembered that LGBTIQ is not a gender, as lesbians can be both female and gays can be male. "other" or "intersex" should be used as the third answer in such cases. |
‘LGBTIQ’ has been replaced with ‘other’ in Table 1 (line 156) and in the preceding text (line 134).In the questionnaire the response options were: Male, Female and Other (specify). |
|
Line 145 Table 1. The authors write in the Data analysis that they use age as an independent variable, but there is no age distribution of the respondents on demographics in the table. Was this intentional? |
The age distribution of the respondents has now been included in Table 1 (line 156). |
|
Line 148: wrongly summed results should be 57.8%, not 27.8% |
The summed results are now correctly indicated as 47.8% (not 57.8%) (line 159). |
|
Line 16-161: Is this a good place for this sentence? Perhaps it should be under the table or in Table 1? |
We deleted the sentence and now include the vaccine literacy results in Table 3 (line 180). |
|
Line 167-168: it will be better "for three out of ten people" (29.2%) |
This change has been made (line 177). |